# Targeted recruitment of USP15 enhances CTLA4 surface levels and restricts its degradation

Francesca Querques[1],*, Victoria Ciampani[1],*, Pei Yee Tey[1], Victoria D Kutilek[2], Elma Kadic[2], Michael J Clague[1], Sylvie Urbé[1]

**Induced protein proximity offers powerful new routes to modulate protein fate. Whereas proteolysis-targeting chimeras (PROTACs) promote degradation through E3 ligase recruitment, the converse principle, targeted protein stabilisation or enhancement via deubiquitylase (DUB) recruitment, is only beginning to emerge. The immune checkpoint receptor CTLA4, whose deficiency causes severe autoimmunity, undergoes rapid ubiquitin-dependent lysosomal degradation, making it one of the most short-lived transmembrane proteins. Using an inducible "RapTag" system, which brings together tagged proteins through rapalog-mediated FKBP–FRB dimerisation, we show that enforced proximity to the broad-specificity DUB USP15 markedly increases total and cell surface CTLA4 levels. Controlled expression of WT or catalytically inactive USP15 in isogenic cell lines revealed a clear requirement for DUB activity. The elevation of CTLA4 at the plasma membrane exceeded that of the total cellular pool, consistent with a diversion from ubiquitin-driven lysosomal sorting towards recycling. This easily adaptable platform enables systematic testing of DUB–substrate combinations that informs rational Enhancement Targeting Chimera (ENTAC) design for downstream drug discovery efforts and targeted protein rescue in therapeutic contexts.**

## Introduction

Creating neo-proximity by chemical approaches offers vast opportunities to influence protein function. At the vanguard of this strategy has been the development of molecular glue degraders and the related Proteolysis Targeting Chimera (PROTAC) molecules (1). Here, an E3 ubiquitin ligase is recruited to a target protein of interest (POI) to elicit its degradation. Despite considerable efforts, the repertoire of E3s that have generated effective PROTACs remains limited, with cereblon and VHL highly favoured. Even then,

these can exhibit a wide range of efficacies depending on the protein target. PROTACs promise to open up the druggable landscape as they require only a binding rather than inhibitory activity towards the POI. However, because of their "event-driven" mode of action, they can be configured with known inhibitor molecules to markedly reduce the effective dose (2, 3).

Ubiquitylation is a reversible post-translational modification mediated by the action of the deubiquitylase (DUBs) family of enzymes (4). A major physiological function of the DUBs is to stabilise proteins, such as c-Myc by USP28, p53 by USP7 and pericentrosomal proteins by USP9X (5, 6, 7). Attention is now turning to leveraging DUB activity by induced proximity, for the stabilisation of selected targets that are defective or limiting in disease settings. This opens a new pharmacology for targeted protein enhancement using chimeric molecules (Enhancement Targeting Chimeras, "ENTACs," elsewhere referred to as "DUB-TACs"). Previous enhancement strategies have been limited to allosteric activators and some molecular chaperones. An early proof-of-principle was the observation that an in-frame fusion of the Lys63 ubiquitin chain-specific DUB, AMSH, to the C-terminus of the epidermal growth factor receptor (EGFR) could deflect the activated receptor from lysosomal degradation (8, 9). A similar approach has been used to generate a library of so called yeast anti-ligases by fusion of E3 enzymes with DUBs (10).

In the absence of non-inhibitory DUB ligand chemistry, the first steps along the ENTAC development pathway involve testing the induced pairing of a POI with a selected DUB. An engineered DUB (USP21) fused to a GFP nanobody was used to target mutant forms of the Cystic Fibrosis Transmembrane Regulator (CFTR) tagged with YFP. This approach partially restores its plasma membrane pool by rescue from Endoplasmic Reticulum Associated Degradation (ERAD) (11). Conceptually, ENTACs are likely to be most effective on proteins that are highly regulated by ubiquitylation, which can impact stability, localisation or function. From the point of view of initial assay development, this is most easily established for proteins that have a half-life of less than 8 h (12, 13, 14). Future applications may incorporate the many proteins that show context-dependent fast degradation kinetics. Examples include

[1]Biochemistry, Cell and Systems Biology, Institute of Systems, Molecular and Integrative Biology, University of Liverpool, Liverpool, UK   [2]Entact Bio, Watertown, MA, USA

Correspondence: clague@liverpool.ac.uk; urbe@liverpool.ac.uk
Victoria D Kutilek's present address is Merck Research Laboratories, Boston, MA, USA
*Francesca Querques and Victoria Ciampani contributed equally to this work

Anaphase-promoting complex (APC) substrates on exit of mitosis, receptor tyrosine kinases on ligand stimulation and metabolic enzymes subject to negative feedback control such as glutamine synthetase (15, 16, 17).

Recently, work from our laboratory has detailed the trafficking itinerary of the immune checkpoint receptor CTLA4 (18). CTLA4 deficiency is a rare primary immune syndrome that leads to excessive T cell activation, contributing to autoimmunity and lymphoproliferation (19). Among proteins with a transmembrane domain, CTLA4 is one of the least stable, exhibiting a half-life of less than 1 h in epithelial cells and 2 h in activated T cells (18, 20, 21, 22). Whereas its function requires plasma membrane presentation, its steady-state distribution mostly displays an intracellular localisation. It is efficiently removed from the plasma membrane and then sorted for lysosomal degradation by a complex ubiquitin signal of mixed linkage types (18).

Here we have developed a platform to rigorously test DUB-POI pairings. In this study, we have used the relatively abundant DUB USP15, which shows broad ubiquitin chain-linkage selectivity, in combination with CTLA4 as a test case (23, 24). The system, which we have termed "RapTag," uses tags on each partner that allow induced interaction through application of Rapalog binding to FK506-binding protein (FKBP) and the FKBP12-rapamycin binding (FRB) domain of mTOR. It also incorporates the inducible expression of DUBs in stable Flp-In cell lines, which allows for exact comparison between WT and catalytically inactive enzymes. We find that induced proximity with USP15 elevates cellular CTLA4 levels. This effect is most profound towards the cell surface pool, which supports a mechanism of reversal of ubiquitin-dependent lysosomal sorting signals. The testing of DUB-POI pairings, in a systematic manner, will provide a means to triage combinations in advance of expensive chemical development campaigns.

## Results

### Design of the RapTag platform

To establish a streamlined platform for efficient testing of DUB-POI pairings, we adopted the use of FKBP- and FRB-tags for chemically induced interaction mediated by a rapamycin analogue. In our system, DUBs are fused to FKBP, whereas the POI is tagged with the FRB-T2098L mutant (referred to as FRB*), which allows for selective interaction with a small membrane-permeant ligand, the A/C heterodimeriser, C16-(S)-7-methylindolerapamycin, also referred to as AP21967/C16-AiRap (Fig 1A) (25). Our DUB and POI modules feature additional N- or C-terminal FLAG- and HA-tags, respectively, for easy detection by Western blotting and immunofluorescence microscopy (Fig 1B). DUBs and POIs can be swapped using restriction enzymes or Gibson assembly and all DUBs are paired with a catalytically inactive counterpart generated by mutation of the active site cysteine (Cys>Ser; CS). We elected to generate isogenic stable HEK293 T-REx Flp-In cell lines, wherein WT or inactive DUBs can be expressed in a doxycycline (Dox) inducible fashion upon integration at a unique genomic locus, alongside a puromycin resistance cassette. This ensures equal, titratable expression levels of any chosen DUB alongside its inactive control, whereas the POI is introduced via transient transfection (Fig 1C). The schedule can be adjusted depending on the specific DUB-POI pairing, but our optimised protocol combines DUB induction and POI transfection in one step, followed 18 h later by application of the heterodimeriser (A/C) for a 24 h period before assay readout (Fig 1D). We aim to observe an increase in protein abundance, which is contingent on DUB induction (+Dox), DUB activity (WT, not CS), and the heterodimeriser (+A/C).

### Benchmarking USP15 RapTag cell lines

For this proof-of-principle study, we chose USP15 as both a cytoplasmic and nuclear distributed DUB to prototype this platform. USP15 is widely expressed at relatively high levels and is known to have broad ubiquitin chain-linkage type selectivity (23, 24). We first established and characterised FLAG-FKBP-(FF)-USP15 WT and inactive USP15 C269S (CS) Flp-In cell pools. A time-course of Dox treatment shows that WT and CS USP15 are expressed at equal levels with respective overexpression compared with endogenous reaching from ~1.4 to 2.5-fold after 4 h to ~21.5 to 23.5-fold after 18 h induction (Fig 2A). For RapTag experiments scheduled according to the protocol shown in Fig 1D, overexpression of FF-USP15 WT or CS reach ~40 to 50-fold the levels of endogenous USP15 at the point of lysis (Fig 2B). Immunofluorescence microscopy confirmed that (i) most of the cells in the pool uniformly expressed the transgene after 24 h of induction and (ii) that FF-USP15 WT and CS mutant localise to both cytosolic and nucleoplasmic compartments as previously reported for endogenous USP15 (Fig 2C) (26). Lastly, we incubated cell lysates derived from FF-USP15 expressing cells with a reactive ubiquitin probe (Ubiquitin-Propargylamide, Ub-PA), which binds to WT but not CS mutant FF-USP15, producing a characteristic upshift in SDS–PAGE (Fig 2D) (27).

Having validated these isogenic WT and inactive mutant USP15 RapTag cell lines, we first established the functionality of our platform on a well-characterised short-lived nuclear protein, c-Myc. Using our optimised protocol, we observed a 1.5 to 2-fold increase in HA-FRB*-c-Myc levels only upon Dox induction in FF-USP15 WT but not in CS expressing cells and only if treated for 24 h with the heterodimeriser A/C (Fig S1A and B).

### CTLA4 degradation can be rescued by targeted recruitment of USP15

We next turned to CTLA4 as an example of a short-lived transmembrane protein, which is constitutively degraded via ubiquitin-dependent trafficking to the lysosome (18). We tagged the cytoplasmically exposed, ubiquitylated C-terminus of CTLA4, with FRB* and HA, to ensure accessibility to USP15 (Fig 3A). We then transiently transfected this construct, CTLA4-FRB*-HA (CTLA4-FH), into FF-USP15 HEK293 cells (-Dox) and then inhibited translation with cycloheximide (CHX) to estimate protein half-life. Note that these cells do not express endogenous CTLA4. We found that >40% of CTLA4-FH is lost within 4 h of CHX treatment (Fig 3B). Conversely, treating the transfected cells with a ubiquitin E1 inhibitor, TAK-243,

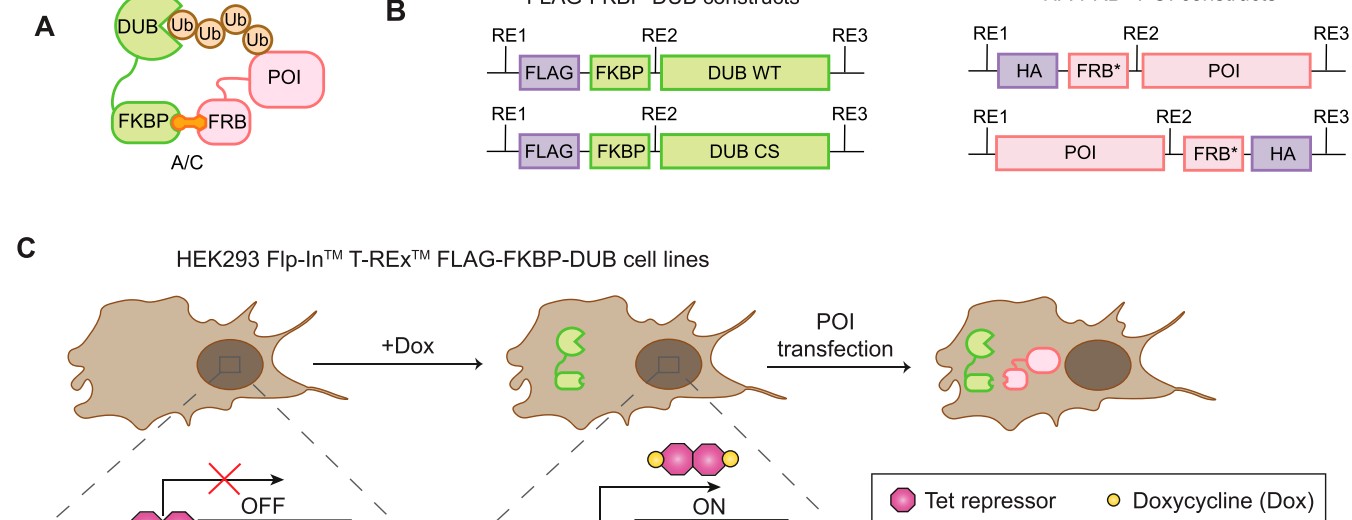

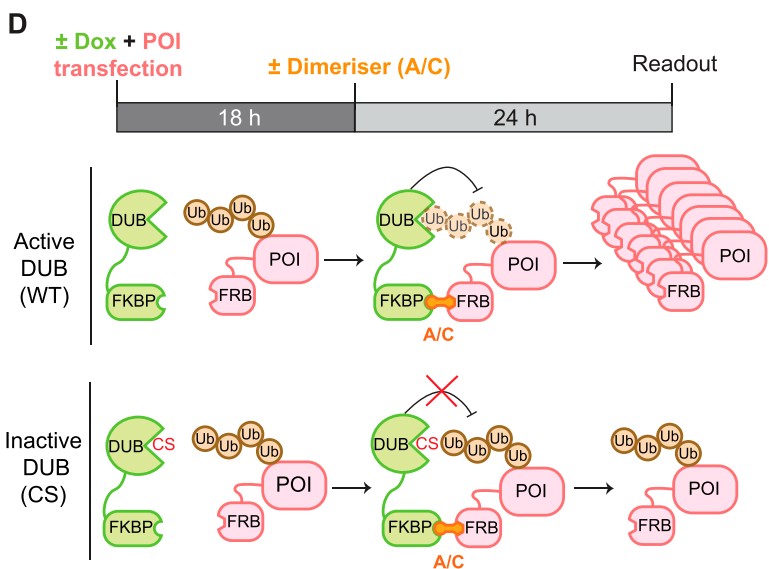

**Figure 1. Design of the "RapTag" platform for ENTAC development.**
**(A)** Schematic illustration of the RapTag system. The A/C heterodimeriser is used to induce artificial proximity between a protein of interest (POI) fused to the FRB-T2098L (FRB*) and a DUB tagged with FKBP12 (FKBP). **(B)** Schematics of WT and catalytically inactive (C>S mutation; CS) DUB and POI constructs used in the RapTag system. Restriction sites (RE) allow swapping of the DUB and POI modules for target validation and DUB/POI pairing; FLAG- and HA-tags are used for easy detection. **(C)** Cartoon of isogenic cell lines generated for doxycycline (Dox)-inducible, equal expression of WT and mutant DUBs. **(D)** Schematic of the RapTag experimental protocol. DUB expression is induced with Dox and the POI is transfected into the cells. After 18 h, the medium is exchanged and the A/C dimeriser (no Dox) is added for another 24 h before harvesting for a variety of readouts (Western blotting, Immunofluorescence, HiBiT assay). FRB, FKBP-rapamycin binding domain (FRB) of the mammalian target of rapamycin (mTOR) kinase; FKBP12, rapamycin binding protein FK506-binding protein; FRT, Flp recombination target.

and to a lesser degree, the lysosomal v-ATPase inhibitor concanamycin A, increased CTLA4-FH protein levels, whereas inhibition of the proteasome with epoxomicin was without impact (Fig 3B). Induction of FF-USP15 WT, but not the CS mutant, imparts >2-fold increase in CTLA4-FH levels that is contingent on the A/C heterodimeriser (Fig 3C and D). Reciprocal pulldown experiments of CTLA4-FH and FF-USP15 under these conditions demonstrate that ternary complex formation is comparable in WT and CS mutant cells (Fig 3E and F; note increased levels of CTLA4-FH in WT [+Dox, +A/C] input samples). Together, these experiments provide strong evidence that CTLA4 is stabilised because of chemically induced heterodimerisation with, and consequential deubiquitylation by, USP15.

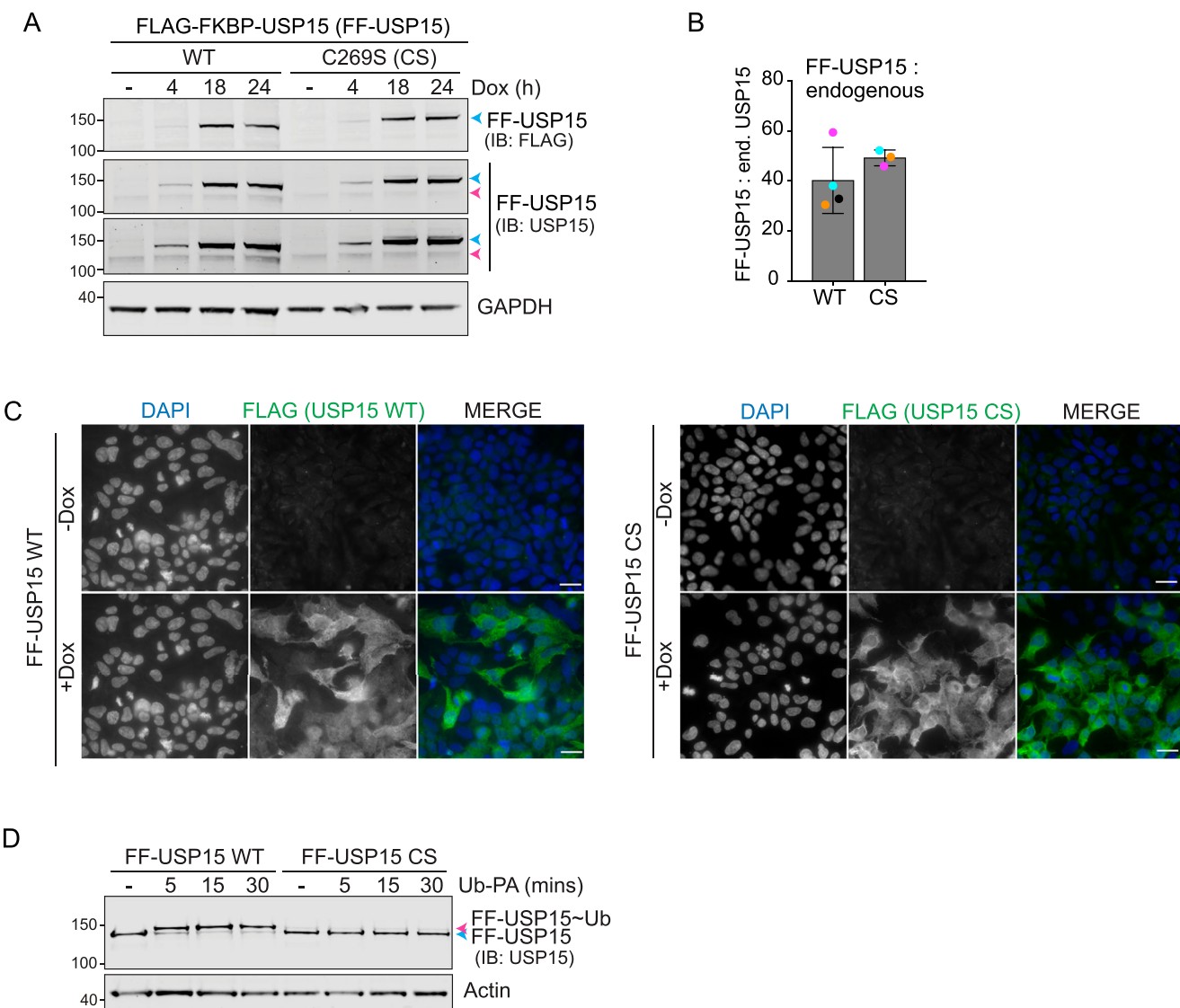

**Figure 2. Validation of the HEK293 Flp-In T-REx FLAG-FKBP-USP15 (FF-USP15) cell lines.**
**(A)** Representative Western blot of samples from HEK293 Flp-In T-REx FLAG-FKBP-USP15 (FF-USP15) WT and catalytically inactive FF-USP15-C269S (CS) cells treated with doxycycline (Dox; 0.1 µg/ml) for indicated times. Blue arrow: FF-USP15; Magenta arrow: endogenous USP15. Note that CS mutant USP15 presents with an additional higher molecular weight band corresponding to a mono-ubiquitylated species. **(B)** Quantification of FF-USP15 levels normalised to endogenous USP15 at the point of lysis of a typical RapTag experiment (see Fig 1D, 42 h after Dox induction). Error bars show SD from four (WT) or three (CS) independent colour-coded experiments.
**(C)** Representative images of FF-USP15 cells induced for 24 h with Dox before PFA fixation and processing for immunofluorescence microscopy with anti-Flag (Alexa Fluor 488; green) using a Nikon Ti Eclipse fluorescence microscope (60X Oil N.A. 1.4 Objective). Nuclear counterstain (DAPI) is shown in blue in the merged image. Scale bar: 20 µm. **(D)** Lysates from FF-USP15 WT and CS expressing cells (24 h Dox) were incubated for indicated times at 37°C with the DUB-probe at a 1:100 (wt/wt) ratio of Ub-PA to total protein and processed for Western blotting with anti-USP15. Blue and magenta arrowheads indicate unbound and Ub-PA bound (~Ub) FF-USP15, respectively. IB, immunoblot.
Source data are available for this figure.

## Induced recruitment of USP15 stabilises CTLA4 at the cell surface

In principle, USP15 could be recruited to and stabilise CTLA4 at any stage of its intracellular trafficking itinerary from the ER via the Golgi to the plasma membrane and inwards up to the sorting endosome. At this point, it is removed from the cytoplasmic interface and packaged into multivesicular bodies (MVBs) for delivery to lysosomes (28). Ubiquitylation is known to act as a signal for internalisation of receptors from the cell surface and for lysosome-directed sorting at the endosome (29). We thus wondered whether the increase in total CTLA4-FH protein levels observed by Western blotting would translate into an increase in the cell surface pool. To this end, we engineered our CTLA4 construct to include an 11–amino acid N-terminal HiBiT peptide exposed to the extracellular medium and thus accessible for complementation with LgBiT included in the extracellular HiBiT assay kit (Fig 4A and B). This reconstitutes a functional nanoluciferase that can be used to quantify steady-state cell surface levels of HiBiT-CTLA4-FH. We

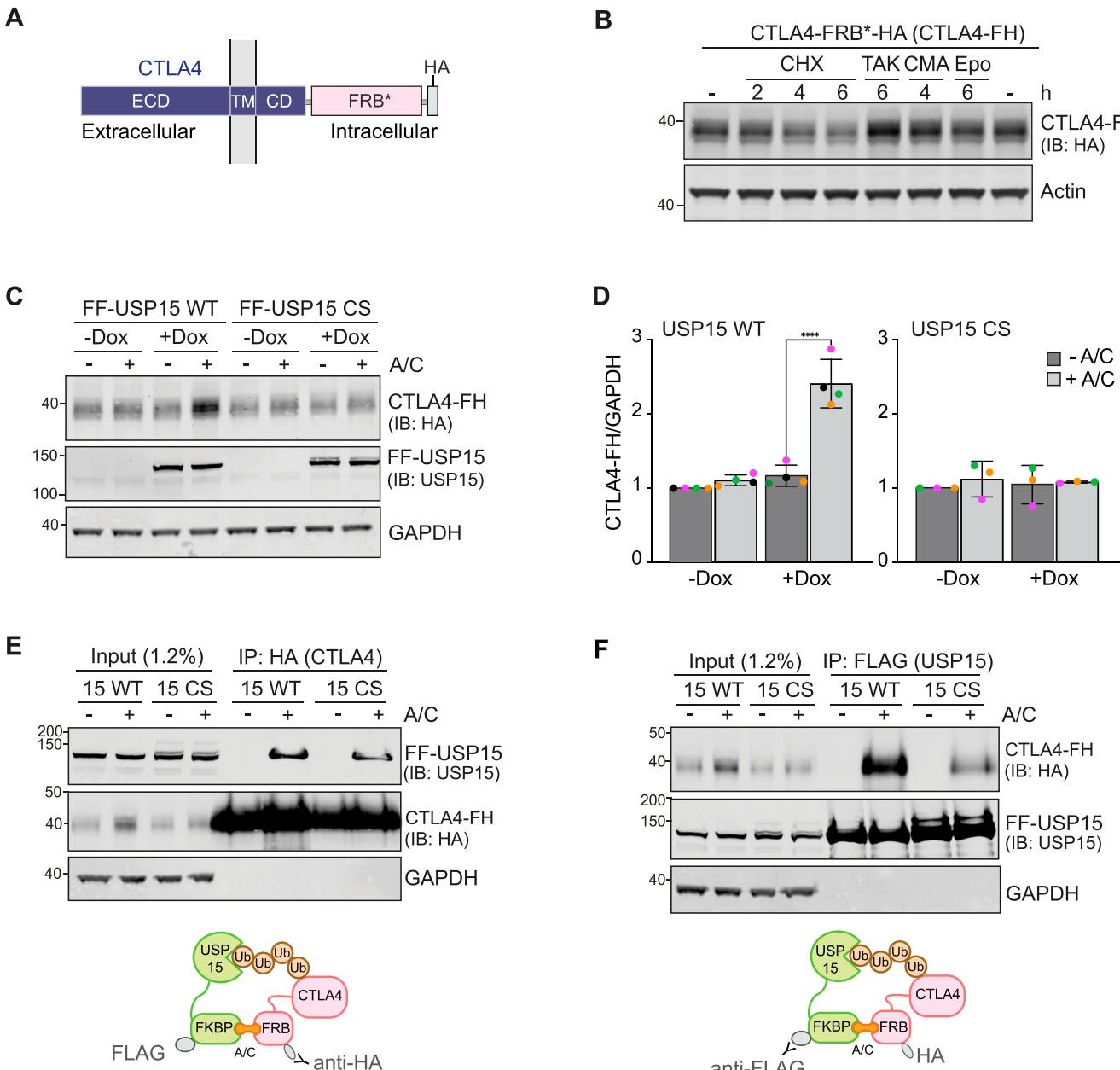

**Figure 3. CTLA4 is a model integral membrane ENTAC substrate that can be rescued by USP15.**
**(A)** Schematic of domain structure and membrane topology of the CTLA4-FRB*-HA (CTLA4-FH) construct. **(B)** Representative Western blot (n = 3) of samples from non-induced FF-USP15 WT cells treated with cycloheximide (CHX, 100 µg/ml), TAK-243 (TAK, 1 µM), concanamycin (CMA, 100 nM), or epoxomicin (Epo, 1 µM) for indicated times before lysis. **(C)** Representative Western blot of HEK293 Flp-In T-REx FLAG-FKBP-USP15 WT (FF-USP15 WT) and C269S (FF-USP15 CS) cells, transfected with CTLA4-FH and treated with or without doxycycline (Dox; 0.1 µg/ml) for 18 h, then treated ± A/C (500 nM) for 24 h before lysis (see Fig 1D). **(D)** Quantification of data represented in (C). Error bars represent SD from four (WT) or three (CS) independent colour-coded experiments. Statistical significance was determined using a two-way ANOVA with uncorrected Fisher's LSD. ****$P < 0.0001$. **(E, F)** Representative Western blots (n = 2) of co-immunoprecipitation experiments to assess ternary complex formation upon A/C treatment. **(E, F)** FF-USP15 WT and CS cells were transfected, induced with Dox and treated ± A/C as in (C) before lysis and immunoprecipitation (IP) with anti-HA (E) or anti-FLAG (F) magnetic beads. ECD, extracellular domain; TM, transmembrane domain; CD, cytoplasmic domain; FRB*, FRB-T2098L; IB, immunoblot. Source data are available for this figure.

first verified that this new construct can still be stabilised upon doxycycline induction of FF-USP15 WT, but not CS, in the presence of A/C (Fig S2A–C). Turning to the highly sensitive bioluminescence-based HiBiT assay, the ~1.5-fold stabilisation seen by Western blotting (Fig S2A–C), transforms into a >4-fold increase in cell surface CTLA4 (Figs 4C and S2D). Using an anti-HiBiT antibody on non-permeabilised cells co-stained with fluorescent wheat germ agglutinin (WGA) (labelling plasma membrane glycoproteins), we could also visualise this specific accumulation at the cell surface (Fig 4D). Collectively, these experiments

**A**

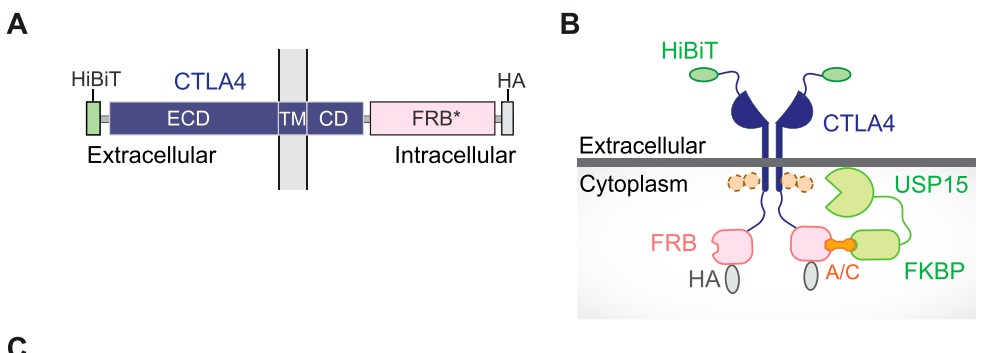

**B**

**C**

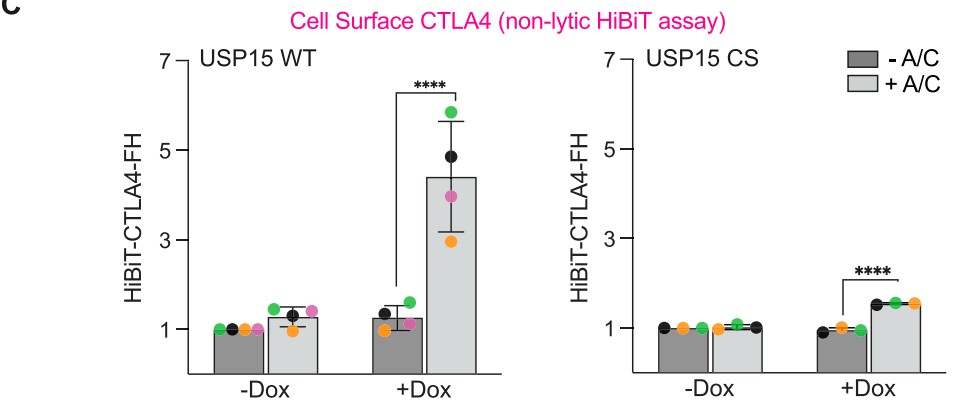

Cell Surface CTLA4 (non-lytic HiBiT assay)

**Figure 4. Chemically induced recruitment of USP15 stabilises CTLA4 at the cell surface.**
**(A, B)** Schematic of domain structure and membrane topology of the HiBiT-CTLA4-FRB*-HA (HiBiT-CTLA4-FH) construct.
**(C)** Extracellular (non-lytic) HiBiT assay performed on FF-USP15 WT and CS cells transfected with HiBiT-CTLA4-FH and treated with or without doxycycline (Dox; 0.1 μg/ml) for 18 h, then treated for 24 h ± A/C (500 nM). Error bars represent SD from four (WT) or three (CS) independent colour-coded experiments each performed in triplicate. Statistical significance was determined using a two-way ANOVA with uncorrected Fisher's LSD. ****$P$ < 0.0001. See also Fig S2D for corresponding cell viability data.
**(D)** Representative images of FF-USP15 WT and CS cells treated as in (C) (all + Dox, ± A/C), fixed and stained without permeabilisation with anti-HiBiT (Alexa Fluor 488, green) and wheat germ agglutinin (WGA) coupled to Alex Fluor 647 (magenta) before imaging using an LSM900 Airyscan confocal microscope (63x Oil objective). Nuclear counterstain (DAPI) is shown in blue in the merged image. Scale bar is 20 μm.

**D**

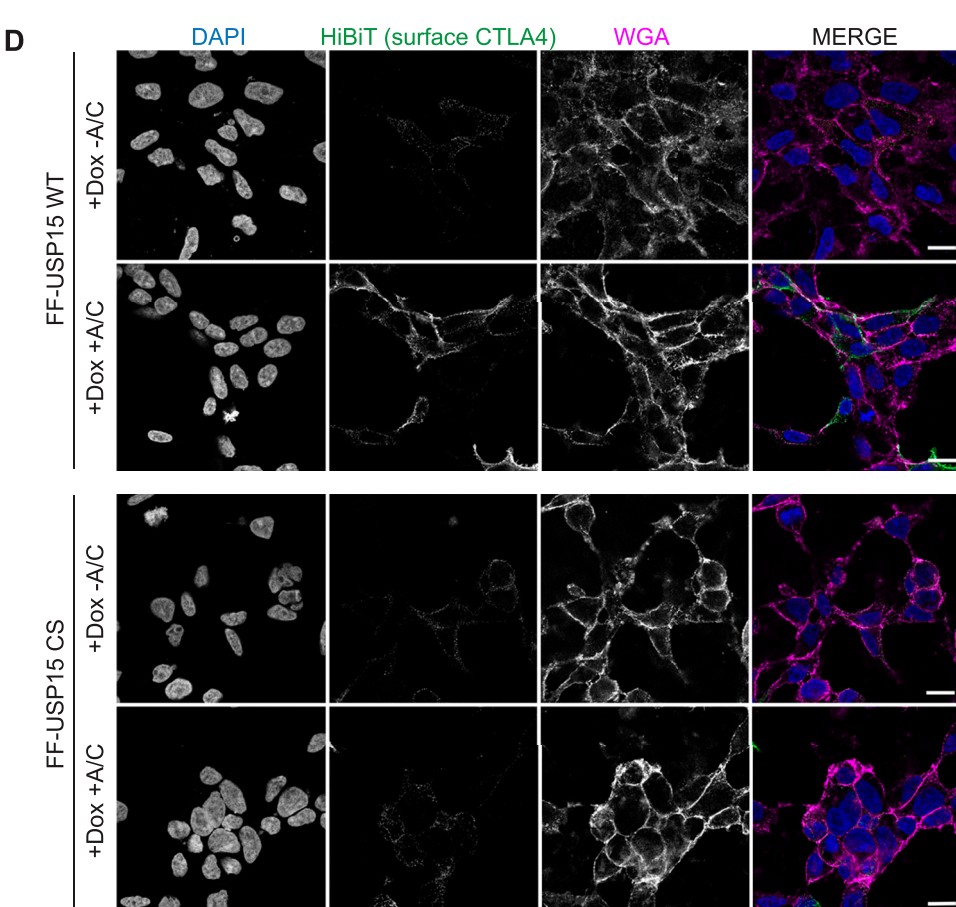

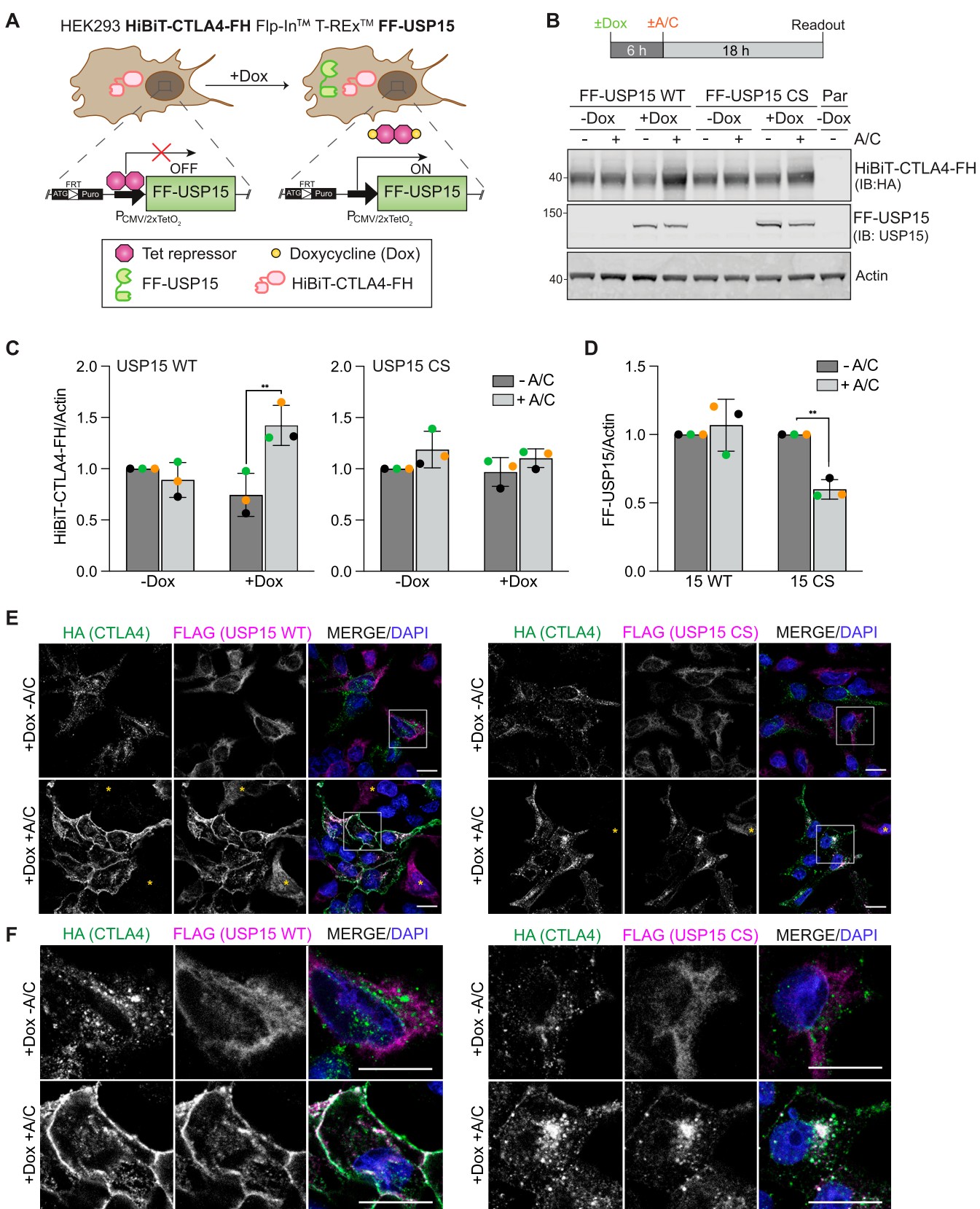

**Figure 5. Distinct fates of active (WT) and inactive (CS) FF-USP15:CTLA4-FH complexes.**
**(A)** Cartoon of second generation Dox-inducible FF-USP15 WT and C269S (CS) HEK293 Flp-In T-REx cells constitutively expressing HiBiT-CTLA4-FH. **(B)** Treatment schedule and representative Western blot of FF-USP15 and CS cell pools stably expressing HiBiT-CTLA4-FH treated with or without doxycycline (Dox; 0.1 μg/ml) for 6 h,

demonstrate a clear increase in the cell surface pool of CTLA4, which is otherwise barely detectable under all three control conditions (no Dox = no DUB, no A/C = no recruitment, USP15 CS +Dox, +A/C = no active DUB).

The transient transfection of the POI offers rapid screening capability, but it is challenging to exactly control protein expression levels between experiments. We, therefore, decided to establish a second set of RapTag cell lines, stably expressing HiBiT-CTLA4-FH alongside the Dox-inducible FF-USP15 (Fig 5A). These remained sensitive to E1 ubiquitin-activating enzyme and lysosomal pathway inhibitors (Fig S3A and B). Applying a shortened Dox-induction schedule of 6 h followed by 18 h of A/C treatment, resulted in a ~2-fold stabilisation (compared with –A/C) of HiBiT-CTLA4-FH in the FF-USP15 WT expressing cells (Fig 5B and C). Catalytically inactive USP15 CS was unable to rescue CTLA4, but we also noticed that its own steady-state levels were reduced by half (Fig 5D). Fluorescence microscopy confirmed the striking A/C-dependent accumulation of CTLA4 at the plasma membrane of WT, but not CS mutant, co-expressing USP15 cells (Figs 5E and F and S3C). A/C treatment also caused a dramatic relocalisation of cytosolic WT FF-USP15 to the plasma membrane where it colocalised with HiBiT-CTLA4-FH (Fig 5E and F). In contrast, inactive FF-USP15 CS redistributed primarily to intracellular punctate structures that were positive for HiBiT-CTLA4-FH (Fig 5E and F). Fluorescence microscopy of the cell pool also revealed cells that failed to express one or other cognate interactor. In the small subpopulation of cells devoid of CTLA4 expression, both WT and inactive USP15 retained a cytoplasmic and nuclear distribution ± A/C (Fig 5E, yellow asterisks). In agreement with our Western blot results (Fig 5D), we noticed an overall decrease in signal intensity for FF-USP15 CS in cells treated with A/C. We reasoned that this may reflect bystander degradation because of complex formation with CTLA4 and subsequent rapid shuttling of ubiquitylated CTLA4-USP15 CS heterodimers to the lysosome. Consistent with this hypothesis, inclusion of the lysosomal (v-ATPase) inhibitor concanamycin A resulted in a marked rescue in the FF-USP15 CS protein levels and signal intensity (Fig S3D–F).

Importantly, this second iteration of the RapTag platform, incorporating HiBiT-tag technology in a stable cell line, allowed us to directly compare gains in cell surface versus total CTLA4 levels in a very reproducible and highly sensitive manner. These experiments demonstrated that the enhancement of CTLA4 cell surface levels outstripped the increase in total CTLA4 by ~2-fold, consistent with the net relocalisation of CTLA4 to the plasma membrane that we observed by immunofluorescence microscopy (Fig 6A and B). We note that the increased robustness and superior sensitivity of the HiBiT-based assay reveal a small but significant increase in both total and cell surface CTLA4 also in cells expressing inactive USP15, most likely reflecting a steric impairment on the recruitment of trafficking adapters or the incorporation of CTLA4 into endocytic carriers.

As a final validation of our platform, we conducted a pulldown experiment using TUBEs to enrich ubiquitylated proteins and probed for CTLA4. Under steady-state conditions, only a small fraction of CTLA4 is ubiquitylated, which is rapidly degraded in the lysosome and challenging to quantify. We recently showed that this pool can be dramatically increased by depleting the endosomal DUB USP8 (18). Under these conditions, we observed a clear 2-fold decrease in the relative fraction of ubiquitylated CTLA4 species (Ub-CTLA4/input CTLA4) in WT USP15 expressing cells treated for the last 20 h with A/C (Fig 6C and D). In contrast, in USP15 CS mutant expressing cells, ubiquitylated CTLA4 levels, if anything, increased with A/C inclusion. Altogether, our data support a model wherein chemically induced recruitment of USP15 to CTLA4 not only increases its stability but also promotes its recycling back to the plasma membrane at the expense of lysosomal degradation (Fig 6E).

## Discussion

The development of ENTACs offers multiple therapeutic opportunities to correct for reduced protein function or stability. Whereas chemical ligands for VHL and cereblon have enabled the development of PROTAC drugs, the optimal ligands for DUBs have likely yet to be identified. Claims for protein stabilisation have been made for ENTAC/DUBTAC rescue of POIs using chimeric DUB inhibitor molecules targeting USP1, USP7 and USP28 (30, 31, 32). Without evidence to the contrary, we feel it is likely that at least some of these observations will simply reflect steric effects of the recruitment of a DUB to the POI rather than deubiquitylation activity. A covalent ligand for OTUB1 residing outside its catalytic site has been identified, which preserves DUB activity (33). This has also been leveraged to affect protein stabilisation of several proteins, but a mode of action is not clearly established (34, 35). OTUB1 is not simply a deubiquitylase; it may also suppress ubiquitylation through binding to certain E2 enzymes and inhibiting their discharge of ubiquitin onto a substrate (36). Here we introduce an isogenic cell platform that allows direct comparison between WT and catalytic mutant USP15 for stabilisation of POIs. USP15 and its catalytically inactive counterpart are both highly

then treated for 18 h ± A/C (500 nM) before lysis. Par: parental uninduced (-Dox) FF-USP15 WT cells. **(C, D)** Quantification of data represented in (B). Error bars show SD from three independent colour-coded experiments. Statistical significance was determined using two-way ANOVA with uncorrected Fisher's LSD. **P < 0.01. **(E)** Representative immunofluorescence microscopy images of FF-USP15 and CS cell pools stably expressing HiBiT-CTLA4-FH, treated with doxycycline (Dox; 0.1 µg/ml) for 6 h, then treated for 24 h with A/C (500 nM) before fixation with methanol and staining with anti-HA (Alexa Fluor 488, green) and anti-FLAG (Alexa Fluor 647, magenta). Nuclear counterstain (DAPI) is shown in blue in the merged image. Images were acquired using a LSM800 confocal microscope (63x Oil objective). Scale bar: 15 µm. Yellow asterisks indicate cells that only express FF-USP15/CS but not CTLA4-FH. **(F)** Enlarged detail of boxed area shown in (E). Scale bar: 15 µm. FRT, Flp Recombination Target; IB, immunoblot.
Source data are available for this figure.

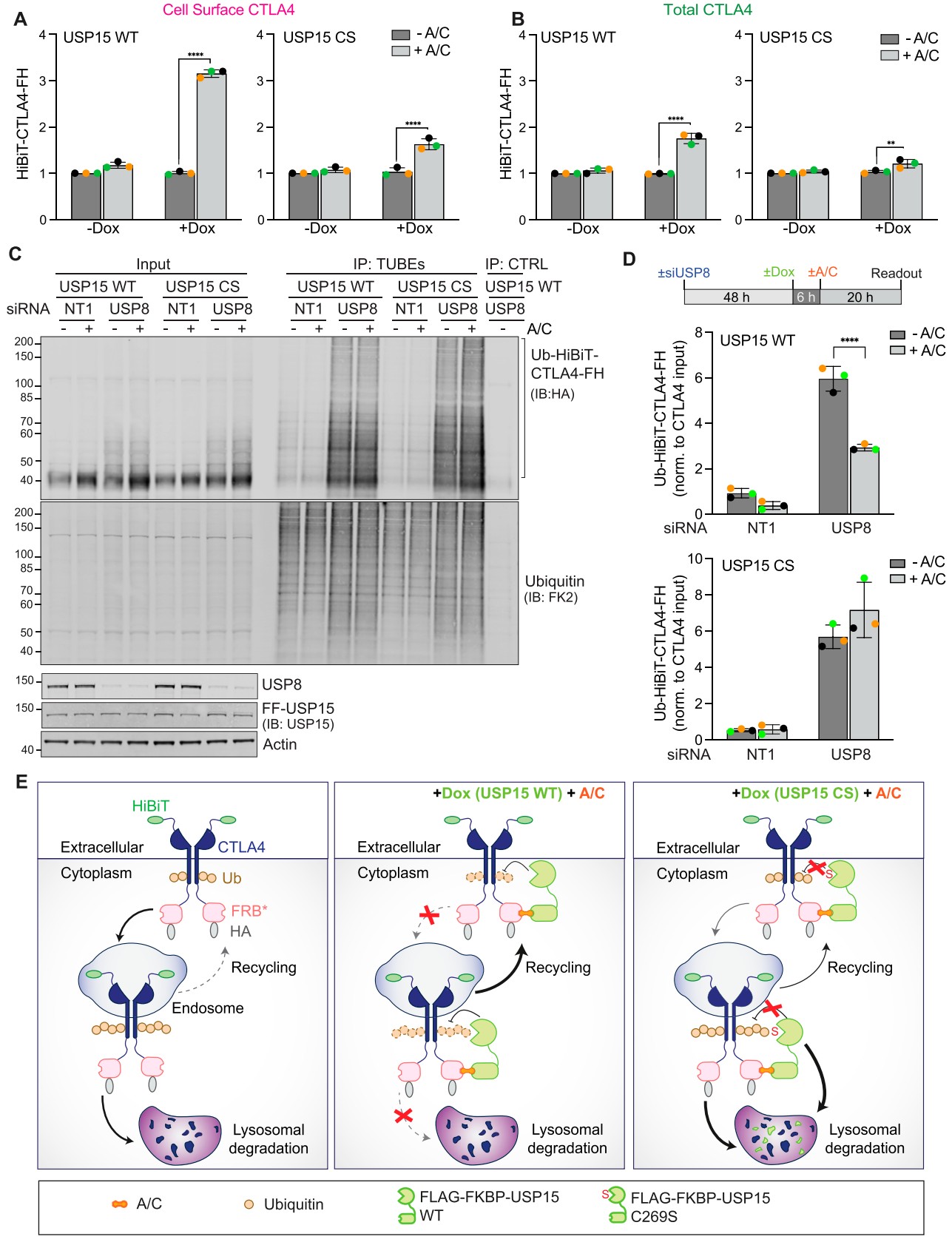

stable proteins under basal conditions. The conserved site of insertion at the Flp-In FRT site ensures equal expression levels and their tight induction by doxycycline allows indirect effects of the A/C dimeriser to be accounted for.

We first established proof-of-principle of straight forward stabilisation of a target using the proteasomal substrate c-Myc, before tackling an integral membrane protein whose turnover is governed by endolysosomal trafficking. Among this class of potential targets, CFTR is a multispanning membrane protein whose clinical significance is well established. Disease-causing mutations generate proteins that are less efficiently folded at the endoplasmic reticulum and triaged for proteasomal degradation through the ERAD pathway (37, 38). The channel is functional if enabled to progress along the biosynthetic pathway to the plasma membrane, but is also less stable at that location, being subject to endocytosis and sorting for lysosomal degradation (39). A proof-of-principle for ENTAC/DUBTAC rescue of CFTR was established with a dual tag system using YFP-CFTR and a GFP/YFP nanobody fused to the catalytic subunit of USP21 (11). Reassuringly, in these experiments, a catalytically inactive mutant of USP21 failed to restore CFTR. USP21 is a highly active DUB, but its low expression levels in cells may preclude the endogenous protein as a realistic ENTAC enabler (24, 40). Chemical ligands for CFTR have also been combined with non-covalent DUB inhibitors to generate "DUB-TACs" albeit with the caveats described above (30, 31, 32, 33, 34). In none of these reports has the endocytic trafficking of CFTR been assessed and rescue is largely assumed to reflect escape from the ERAD proteasomal degradation pathway.

The influence of over-expressed nanobody-tagged DUBs on the itinerary of the KCNQ1-YFP (Kv7.1) channel protein has been explored in more detail (41). Ubiquitin chains on proteins may have multiple chain topologies conferred by different lysine residues used for linkage (42). Many USP DUB family members are promiscuous towards multiple chain linkage types, but some isolated examples from this family and other sub-types of DUB can display stringent chain linkage specificity (4). Shanmugam and colleagues have varied the nanobody-DUB conjugate to interrogate the influence of different chain specificities on discrete membrane transport steps of KCNQ1-YFP. Whereas data are presented for inactive mutants failing to act upon the KCNQ1 associated ubiquitin chains, these controls are not carried over into the trafficking assays. Nevertheless, several intriguing results are noted, including the outsize influence of the Lys11-specific DUB Cezanne (OTU7B) on KCNQ1 abundance relative to the small

fraction of this linkage type typically recorded in cells at steady state (41, 43).

CTLA4 turnover is dominated by ubiquitin-dependent lysosomal degradation, and it is known to acquire a complex ubiquitin chain structure (18). We are using it here to illustrate the utility of the particular platform and to extend the characterisation of the ENTAC approach to this degradation pathway. We first showed that proximal deubiquitylating activity can enhance CTLA4 expression. CTLA4 functions on the cell surface, yet it is distributed across multiple intracellular compartments. The Lys63-specific endosomal DUB, AMSH, has been proposed to deflect some ubiquitylated receptors away from the lysosomal pathway in favour of recycling to the plasma membrane (44, 45). By the same reasoning, if USP15 is opposing the ubiquitin-dependent sorting of CTLA4 into MVBs, then we may observe a preferential increase in the surface pool through enhanced recycling from sorting endosomes. This is unequivocally demonstrated by immunofluorescence microscopy and by luminescence quantification of an extracellular HiBiT tag. Note that each assay includes an uninduced condition (no Dox: no DUB expression) and is accompanied by a control experiment using an isogenically expressed USP15 CS catalytically dead mutant. We believe this to be a necessary control step in establishing DUB-POI pairs and mechanism of action, before chemical ENTAC experiments. Deubiquitylation, as the mechanism of rescue, is difficult to establish for chemical ligands unless a specific inhibitor of the enlisted DUB is available for control experiments. In the case described here, not only is USP15 CS unable to deflect CTLA4 from endolysosomal sorting, it is also itself directly sorted to, and degraded in, the lysosome by virtue of its own association with CTLA4. Thus, there are two clear fates of each partner, dependent on their association and deubiquitylating activity.

ENTACs/DUBTACs are only now emerging into the limelight and their full potential will only be realised when we have a suite of non-inhibitory chemical ligands towards appropriate DUBs. We suggest that the best of these will recruit promiscuous DUBs, which can rescue multiple targets from degradation. The platform we have introduced here provides a way forward to making direct comparisons between DUBs. Engineering other DUBs into the FRT site will provide an enlarged isogenic panel for assessment of relative potency and versatility. The mode of action probably lies somewhere between the event-driven degradation of PROTACs and the occupancy-driven mechanism of many established drugs (46). We propose that relatively abundant DUBs, such as USP15, may be most effective and that short-lived proteins offer

**Figure 6. Induced recruitment of USP15 preferentially stabilises a cell surface pool of CTLA4.**
**(A, B)** Extracellular (A) and lytic (B) HiBiT assay to assess cell surface and total CTLA4 levels, respectively, performed on FF-USP15 and CS cells stably expressing HiBiT-CTLA4-FH treated ± doxycycline (Dox; 0.1 µg/ml) for 6 h, then treated for 18 h ± A/C (500 nM). Error bars show SD from three independent, colour-coded experiments each performed as technical triplicates. Statistical significance was determined using two-way ANOVA with uncorrected Fisher's LSD. ****$P$ < 0.0001 and **$P$ < 0.01. **(C)** FF-USP15 and CS cells stably expressing HiBiT-CTLA4-FH were transfected with either USP8- or non-targeting (NT1) siRNA for 48 h, then treated for 6 h with doxycycline (Dox; 0.1 µg/ml), and supplemented for another 20 h ± A/C (500 nM). Cells were lysed and samples subjected to a pulldown using TUBEs to enrich ubiquitylated proteins, followed by Western blotting with anti-HA. Representative Western blot depicting the Ub-HiBiT-CTLA4-FH signal. **(D)** Treatment schedule and quantification of data represented in (C). Shown is the Ub-HiBiT-CTLA4-FH enriched in the TUBE pulldown normalised to HiBiT-CTLA4-FH in the input samples. Error bars show SD from three independent colour-coded experiments. Statistical significance was determined using two-way ANOVA with uncorrected Fisher's LSD. ****$P$ < 0.0001. **(E)** USP15 RapTag in action. Left panel: At steady state, cell surface CTLA4 is rapidly internalised and sorted in a ubiquitin-dependent manner for degradation in the lysosome. Middle panel: A/C-induced recruitment of active USP15 deubiquitylates CTLA4 and thereby opposes its internalisation whereas also promoting recycling, resulting in a net increase in cell surface CTLA4. Right panel: Inactive USP15 fails to deubiquitylate CTLA4 and is itself destabilised as a bystander by sorting with CTLA4 to the lysosome. IB, immunoblot; CTRL, control beads.
Source data are available for this figure.

straightforward ENTAC targets. There is now a line of sight towards ENTACs designed to treat multiple monogenic diseases associated with loss of function or destabilisation of proteins. The field will be best served by careful and methodical advancement that informs DUB ligand development and target selection.

# Materials and Methods

## Plasmids

Human USP15 WT and C269S (CS) were PCR-amplified with flanking AgeI/NheI sites using pEGFP-USP15 WT and C269S from reference 47 as a template, cloned into pCR4-TOPO (Thermo Fisher Scientific), sequence-verified and subcloned into pcDNA5/FRT/TO FLAG-FKBP Puro (N-terminal Flag tag followed by amino acids 2-108 of FKBP12; amplified from p3E-FKBP-HA no-pA). p3E-FKBP-HA no-pA was a gift from Kryn Stankunas (plasmid # 82598; http://n2t.net/addgene: 82598; RRID:Addgene_82598 (48); Addgene). CTLA4-FRB*-HA (CTLA-FH) was generated by overlap PCR of human CTLA4-HA using pEF5/FRT/CTLA4-HA (18) and FRB-T2098L (referred to as FRB*, encoding amino acids 2,021–2,113 of the FKBP12-rapamycin binding domain of mTOR) from pEGFP-FRB (gift from Klaus Hahn, [plasmid # 25919; Addgene] (49)), cloned into pcDNA3.1 (Thermo Fisher Scientific) and sequence-verified. pcDNA3 HiBiT-CTLA4-FRB*-HA was generated with the Q5 Site-Directed Mutagenesis Kit (NEB) using non-overlapping primers to insert the HiBiT tag between the signal peptide and extracellular domain of CTLA4 into pcDNA3 CTLA4-FRB-HA (forward primer: 5′-CAAGAAGATTAGCGGGAGCTCCGGTGGCTCGAAAGCAATGCACGTGGC-3′; reverse primer: 5′-AACAGCCGCCAGCCGCTCACGCTGCCGCTGCCGCAGAAGACAGGGATGAA-3′) and sequence-verified. c-Myc was amplified from pCMV4a-Flag-c-Myc gift from Hening Lin (plasmid #102625; http://n2t.net/addgene:102625; RRID: Addgene_102625; Addgene). pBluescript was from Stratagene and pOG44 from Thermo Fisher Scientific.

## Generation and maintenance of HEK293 Flp-In T-REx FLAG-FKBP-USP15 WT and C269S cells

To generate doxycycline-inducible cell lines expressing FLAG-FKBP-USP15 WT (FF-USP15 WT) and C269S (FF-USP15 CS), HEK293 Flp-In T-REx host cells (R78007; Thermo Fisher Scientific) were co-transfected with pcDNA5 FRT/TO FLAG-FKBP-USP15 (WT or C269S) and pOG44 at a ratio of 1:9 using GeneJuice (70967; Merck Millipore) according to the manufacturer's instructions. Transfected cells were selected using 0.5 $\mu$g/ml puromycin dihydrochloride (P7255; Sigma-Aldrich) and cultured as pools in DMEM supplemented with GlutaMAX, 10% FBS, 1 x non-essential amino acid (NEAA), 0.5 $\mu$g/ml puromycin and 15 $\mu$g/ml blasticidin S HCl (R21001; Thermo Fisher Scientific) at 37°C and 5% CO$_2$. To generate stable cell lines constitutively expressing HiBiT-CTLA4-FH in the Dox-inducible USP15 background, HEK293 Flp-In T-REx FF-USP15 WT and CS cells were transfected with pcDNA3 HiBiT-CTLA4-FRB*-HA using GeneJuice. Transfected cells were selected using 400 $\mu$g/ml G-418 (4727878001; Sigma-Aldrich Roche) and cultured as pools in DMEM supplemented with GlutaMAX, 10% FBS, 1% NEAA, 400 $\mu$g/ml G-418, 0.5 $\mu$g/ml puromycin, and 15 $\mu$g/ml blasticidin S HCl at 37°C and 5% CO$_2$. Cells were routinely screened for mycoplasma infection.

## Plasmid transfection and RapTag stabilisation protocol

Cells were transfected with the CTLA4 constructs and pBluescript (as carrier) using 3 $\mu$l of Genejuice (70967; Merck Millipore) per total 1 $\mu$g of DNA according to the manufacturer's instructions. Cells were reverse-transfected and induced with 0.1 $\mu$g/ml doxycycline at the time of seeding. The medium was exchanged 18 h after transfection and cells treated with 500 nM A/C or EtOH (vehicle) for the last 24 h before lysis or fixation. For RapTag experiments with the HEK293 Flp-In T-REx FF-USP15 WT and C269S cell lines stably expressing HiBiT-CTLA4-FH, cells were induced for 6 h with 0.1 $\mu$g/ml doxycycline at the time of seeding, before addition of 500 nM A/C for the last 18 h before analysis.

## siRNA interference

Stable HEK293 HiBiT-CTLA4-FH Flp-In T-REx FF-USP15 WT and CS cells were transfected with 40 nM non-targeting (NT1) or USP8 targeting (oligo 1) siRNA using Lipofectamine RNAiMAX (13778030; Invitrogen) according to manufacturer's instructions. After 48 h, cells were induced with 0.1 $\mu$g/ml doxycycline for 6 h and then treated with 500 nM A/C for the last 18–20 h before lysis. siRNA oligonucleotides were obtained from Dharmacon, Horizon Discovery: USP8 oligo 1 (siGENOME, D-005203-02: 5′-UGAAAUACGUGACUGUUUAUU-3′) and non-targeting 1 (NT1) control (ON-TARGET-plus, D-001810-01, 5′-UGGUUUACAUGUCGACUAA-3′) (50).

## Antibodies and reagents

Antibodies and other reagents used were as follows: anti-HA (MMS-101P; WB 1:1,000; Covance), anti-HA (NB600-362; WB 1:10,000; IF 1: 250; Novus Biologicals), anti-HA (3724; WB 1:1,000; Cell Signaling Technology), anti-Actin (66009; WB 1:5,000; Proteintech), anti-HiBiT (N720A; WB 1:1,000; IF 1:1,000; Promega), anti-GAPDH (2118S; WB 1: 1,000; Cell Signaling Technologies), anti-USP15 (A300-923A; WB 1: 1,000; Bethyl Laboratories), anti-FLAG (F3165 or F1804; WB 1:1,000; IF 1:1,000; M2 Sigma-Aldrich), anti-DYKDDDDK (FLAG) Tag (D6W5B) (14793; IF 1:1,000; Cell Signaling Technology), anti-ubiquitin FK2 (PW8810; WB 1:500; Enzo), anti-USP8 antibody (AF7735; WB 1:500; R&D Systems), Wheat Germ Agglutinin (WGA) Alexa Fluor 647 (gift from Shankar Varadarajan, Liverpool), A/C Heterodimeriser (635055, 500 nM unless otherwise indicated; Takara/Clontech), doxycycline (D9891, 0.1 $\mu$g/ml; Sigma-Aldrich), cycloheximide (C7698, 100 $\mu$g/ml; Sigma-Aldrich), concanamycin A (C9705, 100 nM; Sigma-Aldrich), epoxomicin (324800, 1 $\mu$M; Merck Millipore), TAK-243 (S8341, 1 $\mu$M; Selleckchem), and Poly-L-lysine (P4832, solution, 0.01%; Merck Millipore).

## Cell lysis and Western blot analysis

Cultured cells were rinsed twice with PBS and lysed in denaturing SDS lysis buffer (2% wt/vol SDS, 1 mM EDTA and 50 mM NaF) at 110°C, followed by boiling for 10 min with intermittent vortexing.

Protein concentration was determined using the Pierce BCA protein assay according to the manufacturer's instructions. Samples were diluted with 5x SDS-sample buffer (15% wt/vol SDS, 312.5 mM Tris–HCl pH 6.8, 50% glycerol and 16% β-mercaptoethanol) and boiled at 95°C. Proteins were resolved using SDS–PAGE (NuPage gel 4–12%; Invitrogen), transferred to nitrocellulose membrane (10600002; Amersham Protran), stained with Ponceau S staining solution (P7170; Sigma-Aldrich), and blocked in 5% milk (Marvel) in TBST (TBS: 20 mM Tris–Cl, pH 7.6 and 150 mM NaCl, supplemented with 0.1% Tween-20 [10485733; Thermo Fisher Scientific]) before incubation with primary antibodies overnight. Visualization and quantification of Western blots were performed using IRdye 800CW (anti-mouse 926-32212, anti-rabbit 926-32213, and anti-goat 926-32214) and IRdye 680LT (anti-rabbit 92668023 and anti-mouse 926-68022) coupled secondary antibodies and an Odyssey infrared scanner (LI-COR Biosciences). For Western blot quantification, raw signal values were obtained using ImageStudio Lite (Li-COR Biosciences) after background subtraction.

### Immunoprecipitation

For co-immunoprecipitation experiments, cells were washed twice in ice-cold PBS and lysed in NP40 lysis buffer (0.5% NP40, 25 mM Tris pH 7.5, 100 mM NaCl, and 50 mM NaF) supplemented with mammalian protease inhibitor cocktail (P8340; Sigma-Aldrich) and PhosSTOP (49068450001; Roche) for 10 min on ice. Lysates were clarified by centrifugation and protein concentration was determined using the Pierce BCA protein assay according to the manufacturer's instructions. Clarified lysates were incubated with 25 μl of anti-HA magnetic beads (88837; Thermo Fisher Scientific) or 50 μl of anti-DYKDDDDK (FLAG) magnetic beads (A36797; Thermo Fisher Scientific) for 1 h at RT. Beads were washed three times with 0.05% TBST and proteins were eluted in 1x SDS-sample buffer.

### Tandem ubiquitin binding entities (TUBEs) pulldown

Cells were washed twice with ice-cold PBS and lysed in TUBE lysis buffer (50 mM Tris–HCl pH 7.5, 150 mM NaCl, 1 mM EDTA, 1% [wt/vol] NP40, 10% [wt/vol] glycerol) supplemented with mammalian protease inhibitor cocktail (P8340; Sigma-Aldrich), PhosSTOP (4906845001; Roche) and 10 mM NEM (E3876-5G; Sigma-Aldrich). Lysates were incubated with 35 μl magnetic TUBEs (TUBE 2 beads; UM402M; Life sensors) or control beads (UM400M; Life sensors) overnight at 4°C. Beads were washed with 0.1% TBST and proteins were eluted in 1 × SDS-sample buffer.

### Immunofluorescence microscopy

Cells seeded on Poly-L-lysine-coated coverslips were fixed with 4% paraformaldehyde (PFA, AGR1026; Agar Scientific) in PBS. Excess PFA was quenched with 50 mM $NH_4Cl$/PBS and cells were permeabilised with 0.2% Triton X-100 in PBS. Alternatively, cells were just fixed with methanol for 5 min at −20°C. Fixed cells were incubated for 30 min in blocking solution (10% FBS or 3% BSA [First Link, G6767] in PBS), then stained with primary antibodies (30 mins), followed by Alexa Fluor 488-, Alexa Fluor 594-, or Alexa

Fluor 647-coupled secondary antibodies (Invitrogen) for 20 min in blocking buffer (5% FBS or 3% BSA in PBS).

For HiBiT surface labelling, cells were incubated with the HiBiT antibody (1:1,000) in PBS for 10 mins on ice before PFA fixation. For Fig 4D, 0.4 μg/ml WGA Alexa Fluor 647 was then added together with the Alexa Fluor 488-coupled secondary anti-mouse IgG antibody. For Fig S3C, cells were permeabilised (after HiBiT staining) with 0.2% Triton X-100 in PBS and the standard procedure was followed for primary and secondary incubation with anti-FLAG rabbit and Donkey anti-rabbit Alexa Fluor 647 antibodies. Coverslips were mounted onto glass slides using Mowiol (475904; Merck) containing DAPI (D1306, 1:5,000; Invitrogen). Cells were imaged using an LSM800 or LSM900 Airyscan confocal microscope (63x Oil NA 1.4 objective, acquisition software Zen Blue) or a Nikon Ti Eclipse (60x Oil N.A. 1.4 objective, acquisition software NIS Elements). All images were acquired sequentially and processed using Fiji (version 2.1.0) and Adobe Photoshop (version 24.5.0) software.

### NanoGlo HiBiT and CellTiter-Glo assays

Cells were seeded in opaque white 96-well plates (3610; Corning Costar), co-transfected with HiBiT-CTLA4-FH and pBluescript using GeneJuice and induced with 0.1 μg/ml doxycycline for 24 h, followed by treatment with 500 nM A/C for 24 h. Stable HEK293 HiBiT-CTLA4-FH FF-USP15 WT and CS cells were induced with 0.1 μg/ml doxycycline for 6 h, followed by treatment with 500 nM A/C for 18 h. HiBiT assays were performed with the NanoGlo HiBiT extracellular detection system (N2420; Promega) to detect cell surface CTLA4, or the NanoGlo HiBiT Lytic detection system (N3040; Promega) to measure total CTLA4 levels. The CellTiter-Glo (CTG) assay was performed with the CellTiter-Glo Luminescent Cell Viability Assay (G7571; Promega). Plates were read using the Glomax GM3500 Explorer (Promega). Background subtraction was performed using uninduced (no Dox), untransfected cells for the HiBiT assay, or DMEM for CTG. For extracellular HiBiT assays using transient transfection shown in Fig 4C, only experiments with a basal luminescence signal (RLU in −Dox, −A/C sample) of $<2 \times 10^6$ were included in the analysis.

### Activity-based probe assay

Cells were harvested in non-denaturing buffer (50 mM Tris pH7.5, 5 mM $MgCl_2$, 250 mM sucrose, 1 mM DTT, 2 mM ATP) supplemented with PhosStop (4906845001; Roche) on ice and homogenised by progressively passing through 23G, 26G, and 30G needles. Lysates were clarified by centrifugation (20 min, 18,400$g$, 4°C) and protein concentration determined using the Pierce BCA protein assay according to the manufacturer's instructions. Lysates were incubated with Ub-PA (UbiQ-057; UbiQ) at a ratio of 1:100 (wt/wt) Ub-PA to total protein in lysate, for 5, 15, or 30 min at 37°C with shaking at 300 rpm in an Eppendorf Thermomixer (compact). The reaction was stopped by addition of sample buffer and heating at 95°C.

### Statistical analysis

Graphs were plotted using GraphPad Prism10. Statistical significance was determined using two-way ANOVA with uncorrected

Fisher's LSD. *P*-values are represented as \*\*$P < 0.01$, \*\*\*$P < 0.001$, and \*\*\*\*$P < 0.0001$.

## Data Availability

Data are available in the article itself and its supplementary materials.

## Supplementary Information

## Acknowledgements

We thank the Liverpool University Center for Cell Imaging for providing access to instrumentation. Funding for F Querques and V Ciampani to develop the RapTag platform was provided by Entact Bio. PY Tey was supported by the Biotechnology and Biological Sciences Research Council (BBSRC; BB/Y011058/1). MJ Clague is a Royal Society Industry Fellow (INF\R2\212031).

### Author Contributions

F Querques: conceptualization, data curation, formal analysis, validation, investigation, visualization, methodology, and writing—original draft, review, and editing.
V Ciampani: conceptualization, data curation, formal analysis, validation, investigation, visualization, methodology, and writing—original draft, review, and editing.
PY Tey: conceptualization, data curation, formal analysis, investigation, visualization, methodology, and writing—original draft, review, and editing.
VD Kutilek: conceptualization and methodology.
E Kadic: conceptualization, resources, supervision, methodology, project administration, and writing—original draft, review, and editing.
MJ Clague: conceptualization, resources, supervision, funding acquisition, visualization, methodology, project administration, and writing—original draft, review, and editing.
S Urbé: conceptualization, resources, supervision, funding acquisition, visualization, methodology, project administration, and writing—original draft, review, and editing.

### Conflict of Interest Statement

MJ Clague and S Urbé are founders and members of the scientific advisory board of Entact Bio. E Kadic is an employee of Entact Bio.

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
