## [Reviewer comments · Life Science Alliance]

Targeted recruitment of USP15 enhances CTLA4 surface levels and restricts its degradation.

Francesca Querques, Victoria Ciampani, Pei Yee Tey, Victoria Kutilek, Elma Kadic, Michael Clague, and Sylvie Urbe
DOI: <https://doi.org/10.26508/lsa.202503563>

Corresponding author(s): Sylvie Urbe, University of Liverpool and Michael Clague, University of Liverpool

Review Timeline:

Submission Date:	2025-11-07
Editorial Decision:	2025-12-19
Revision Received:	2026-01-05
Accepted:	2026-01-08

Scientific Editor: Tim Fessenden

Transaction Report:

December 19, 2025

RE: Life Science Alliance Manuscript #LSA-2025-03563

Prof. Sylvie Urbe
University of Liverpool
Cellular and Molecular Physiology
University of Liverpool
Crown Street
Liverpool, Merseyside L69 3BX
United Kingdom

Dear Dr. Urbe,

Thank you for submitting your manuscript entitled "Targeted recruitment of USP15 enhances CTLA4 surface levels and restricts its degradation". Your manuscript was evaluated by three experts whose reports are appended below.

As you will see, all reviewers expressed strong enthusiasm for this work and none conveyed significant concerns. Reviewer 1 requested clarification of whether endogenous USP15 impacts these findings, which we invite you to address in the manner of your choice. Reviewers 2 and 3 made suggestions to improve the text, which we encourage you to consider. We would be happy to publish your paper in Life Science Alliance pending these changes and final revisions necessary to meet our formatting guidelines.

- Please be sure that the authorship listing and order is correct.
- Please upload your main manuscript text as an editable doc file.
- Please upload all figure files as individual ones, including the supplementary figure files; all figure legends should only appear in the main manuscript file.
- Please add the X and Bluesky handles of your host institute/organization, as well as your own and/or one of the authors, in our system.
- The "Data Availability" section should be placed after the Materials & Methods section. Please consult our guidelines at <https://www.life-science-alliance.org/manuscript-prep#format>
- Please move your main and supplementary figure legends in the main manuscript text after the references section.
- It is recommended to exclude figures from the manuscript text and upload them separately.
- Our articles do not include a "Significance Statement". Please remove this from the manuscript text.

A. FINAL FILES:

- An editable version of the final text (.DOC or .DOCX) is needed for copyediting (no PDFs).
- High-resolution figure, supplementary figure and video files uploaded as individual files: See our detailed guidelines for preparing your production-ready images, <https://www.life-science-alliance.org/authors>
- Summary blurb (enter in submission system): A short text summarizing in a single sentence the study (max. 200 characters

including spaces). This text is used in conjunction with the titles of papers, hence should be informative and complementary to the title. It should describe the context and significance of the findings for a general readership; it should be written in the present tense and refer to the work in the third person. Author names should not be mentioned.

B. MANUSCRIPT ORGANIZATION AND FORMATTING:

Thank you for your attention to these final processing requirements. Please revise and format the manuscript and upload materials as soon as you are able.

Sincerely,

Reviewer #1 (Comments to the Authors (Required)):

LSA-2025-03563

Ubiquitylation controls the presence of membrane proteins at the cell surface, including the immune checkpoint molecule CTLA-4. In the current study, Querques et al. have developed cells systems to stringently test the possibility of influencing the cell surface expression of CTLA-4 by inducibly recruiting the deubiquitylase USP15 through the use of a membrane permeable small molecule ligand. In a series of experiments the conclusion is reached that this system is suited to monitor the activity of DUBs and to make direct comparisons between these.

The study is extremely well controlled, pleasant to read, and perfectly well illustrated. This work has substantial innovation potential in the field of ubiquitin biology. One point may need some further clarification: To be able to see the effect on ubiquitylation of induced USP15 recruitment onto CTLA-4, endogenous USP15 needed to be depleted (Fig. 6C,D). The experiments in Fig. 1 to 5 were done in the presence of endogenous USP15. It should be addressed (ideally experimentally, otherwise in through appropriate wording) whether and how key outcomes in these figures would be influenced by depletion of endogenous USP15.

Minor point: C16-(S)-7-methylindolerapamycin is typically written with a small "m".

Reviewer #2 (Comments to the Authors (Required)):

Querques and Ciampani et al., show that chemically forcing proximity between the deubiquitylase USP15 and the immune checkpoint receptor CTLA4 can strongly raise CTLA4 abundance (especially at the cell surface), by diverting ubiquitin-

dependent lysosomal sorting towards recycling instead.

The authors demonstrate this using an inducible "RapTag" system that dimerizes FKBP-FRB-tagged partners and lets them directly compare wild-type versus catalytically inactive DUBs in matched cell lines. In well-designed and executed surface assays, a modest total-protein stabilization translates into a much larger gain in surface CTLA4, underscoring a trafficking reroute rather than simple bulk accumulation.

Altogether, the study outlines a mechanistic basis for ENhancement TArgeting Chimeras (ENTACs) as a "protein rescue" counterpart to protein degraders such as PROTACs, which has the potential to re-wire biological pathways and potentially translate to therapeutics.

Moreover, the study introduces a practical platform for triaging DUB-target pairings before committing to full chemical development, an important framework that could accelerate and de-risk new therapeutic efforts. As such, it may serve as a valuable reference point, helping to define a minimum standard of experimental rigour and validation expected for future ENTAC/DUBTAC studies.

Overall, the article is very well written and the data are clear and well presented (e.g. experiments with the HiBiT-engineered CTLA4 construct are particularly elegant). I only have some minor comments below:

1. "event-driven" mode of action is mentioned at the end of the first paragraph of the introduction. It may be useful to reference a review that highlights this new wave of biopharmaceutical discovery (E.g. PMID: 35042991 or PMID: 32296187), as this may be useful for others who are new to the field.
2. Might be worth referring to Fig. 3B after "...epoxomicin was without impact" in page 6. It took me a couple of reads to figure out where these data were.
3. Several candidate applications are discussed for proximity-based DUB-mediated stabilisation (e.g., CFTR, KCNQ1). This raised a broader question about scope. In many settings, proteasomal degradation is protective or necessary (e.g. misfolded, non-functional, or aggregation-prone proteins), and stabilising such species could be undesirable. Could the authors expand on how they envision managing this risk? More specifically, do they anticipate the ENTAC/DUBTAC strategy being more likely to succeed for membrane proteins and other targets that undergo endosomal sorting and recycling, rather than for substrates primarily marked for proteasomal degradation? Any discussion or speculation on target features that may predict success (or failure) would be helpful.

Reviewer #3 (Comments to the Authors (Required)):

This manuscript introduces a well-designed RapTag system that enables inducible proximity between deubiquitylases (DUBs) and their potential substrates using an FKBP-FRB* heterodimerisation strategy. The authors benchmark the platform using USP15 and demonstrate that enforced proximity of USP15 to the immune checkpoint receptor CTLA4 markedly increases CTLA4 stability. Stabilisation is strongly dependent on catalytic activity and on the heterodimeriser, and is accompanied by a substantial elevation of CTLA4 at the plasma membrane. The combination of biochemical, imaging, HiBiT-based quantification, and ubiquitin enrichment assays supports a model in which USP15 recruitment counteracts ubiquitin-dependent lysosomal sorting, thereby promoting CTLA4 recycling.

The study is clearly presented, technically rigorous, and offers a flexible platform for evaluating DUB-substrate interactions with direct implications for the rational design of enhancing chimeras (ENTACs). The work advances an emerging field and has clear conceptual importance for targeted protein rescue strategies.

I find the manuscript strong and suitable for publication after minor revision aimed solely at improving flow and sharpening the central message.

1. Several Results subsections contain rich technical detail, but the overarching conceptual thread, which enforced DUB proximity can redirect protein fate by altering trafficking decisions, occasionally becomes diluted. Can the authors add brief bridging statements at the end or beginning of relevant subsections to help readers follow the logical progression from platform establishment to validation to mechanistic insight to functional outcome.
2. The observation that inactive USP15 accumulates in intracellular puncta and undergoes reduced steady-state expression is interesting but described briefly. Add one clarifying sentence in the Results or Discussion explaining the likely mechanism (e.g., bystander degradation due to complexing with ubiquitylated CTLA4), ensuring the message remains sharp and accessible to non-specialists.

Response to Reviewers

Life Science Alliance Manuscript #LSA-2025-03563

We thank all three reviewers for their support and constructive comments, which we have taken onboard in the revised version of our manuscript.

Point-by-point response to reviewers comments

Reviewer #1

Ubiquitylation controls the presence of membrane proteins at the cell surface, including the immune checkpoint molecule CTLA-4. In the current study, Querques et al. have developed cells systems to stringently test the possibility of influencing the cell surface expression of CTLA-4 by inducibly recruiting the deubiquitylase USP15 through the use of a membrane permeable small molecule ligand. In a series of experiments the conclusion is reached that this system is suited to monitor the activity of DUBs and to make direct comparisons between these.

The study is extremely well controlled, pleasant to read, and perfectly well illustrated. This work has substantial innovation potential in the field of ubiquitin biology. One point may need some further clarification: To be able to see the effect on ubiquitylation of induced USP15 recruitment onto CTLA-4, endogenous USP15 needed to be depleted (Fig. 6C,D). The experiments in Fig. 1 to 5 were done in the presence of endogenous USP15. It should be addressed (ideally experimentally, otherwise in through appropriate wording) whether and how key outcomes in these figures would be influenced by depletion of endogenous USP15.

Response: We believe this reviewer may have misread the relevant Figure. Figure 6C and D, which the reviewer refers to does not show depletion of endogenous USP15 but of the endosomal DUB USP8, which we previously showed suppresses CTLA4 ubiquitylation and endolysosomal trafficking. By depleting USP8, we can augment the ubiquitylation signal to provide us with a larger signal to noise ratio for the ubiquitin immunoblot. In none of the figures do we deplete endogenous USP15, which is not interfering with the readouts we are measuring here as it is lacking the FKBP tag necessary for recruitment via the A/C dimeriser. We made a few minor adjustments to the text on page 8 to avoid any confusion in this matter.

Minor point: C16-(S)-7-methylindolerapamycin is typically written with a small "m".

Response: OK, this has been corrected.

Reviewer #2:

Querques and Ciampani et al., show that chemically forcing proximity between the deubiquitylase USP15 and the immune checkpoint receptor CTLA4 can strongly raise CTLA4 abundance (especially at the cell surface), by diverting ubiquitin-dependent lysosomal sorting towards recycling instead.

The authors demonstrate this using an inducible "RapTag" system that dimerizes FKBP-FRB-tagged partners and lets them directly compare wild-type versus catalytically inactive DUBs in matched cell lines. In well-designed and executed surface assays, a modest total-protein stabilization translates into a much larger gain in surface CTLA4, underscoring a trafficking reroute rather than simple bulk accumulation.

Altogether, the study outlines a mechanistic basis for ENhancement TArgeting Chimeras (ENTACs) as a "protein rescue" counterpart to protein degraders such as PROTACs, which has the potential to re-wire biological pathways and potentially translate to therapeutics.

Moreover, the study introduces a practical platform for triaging DUB-target pairings before committing to full chemical development, an important framework that could accelerate and de-risk new therapeutic efforts. As such, it may serve as a valuable reference point, helping to define a minimum standard of experimental rigour and validation expected for future ENTAC/DUBTAC studies.

Overall, the article is very well written and the data are clear and well presented (e.g. experiments with the HiBiT-engineered CTLA4 construct are particularly elegant). I only have some minor comments below:

1. "event-driven" mode of action is mentioned at the end of the first paragraph of the introduction. It may be useful to reference a review that highlights this new wave of biopharmaceutical discovery (E.g. PMID: 35042991 or PMID: 32296187), as this may be useful for others who are new to the field.

Response: We thank the reviewer for this excellent suggestion.

2. Might be worth referring to Fig. 3B after "...epoxomicin was without impact" in page 6. It took me a couple of reads to figure out where these data were.

Response: Agreed, we have added this call-out.

3. Several candidate applications are discussed for proximity-based DUB-mediated stabilisation (e.g., CFTR, KCNQ1). This raised a broader question about scope. In many settings, proteasomal degradation is protective or necessary (e.g. misfolded, non-functional, or aggregation-prone proteins), and stabilising such species could be undesirable. **Could the authors expand on how they envision managing this risk?** More specifically, do they anticipate the ENTAC/DUBTAC strategy being **more likely to**

succeed for membrane proteins and other targets that undergo endosomal sorting and recycling, rather than for substrates primarily marked for proteasomal degradation? Any discussion or speculation on target features that may predict success (or failure) would be helpful.

Response: Target selection is clearly one highly important consideration for this application that will require careful evaluation as well as experimental validation, for which the system described in this manuscript provides a straight forward plug-and-play type platform. For aggregation-prone or misfolded proteins, eg CFTR mutant, enhanced deubiquitylation may simply provide “more time” for the protein to gain the correct conformation (eg by engaging engaging multiple rounds of ubiquitylation/deubiquitylation cycles). Ultimately, the final proof for a successful ENTAC strategy for therapeutically relevant targets will need to include a functional read-out. An extensive discussion/speculation on target selection and validation, we believe, would be distracting here and better placed in a review-type manuscript.

We do not mean to imply that membrane proteins or other endosomal cargo will necessarily constitute targets with a higher success-rate. However, we do provide a proof-of-principle that such targets can not just be rescued from degradation but functionally enabled by affecting their subcellular localisation. As CTLA4 has an extremely short half-life, this makes it an excellent test case for our RapTag platform. Whilst we have not expanded widely on proteasomal targets, we had shown in Supplementary Figure 1 that the transcription factor c-Myc can likewise be stabilised by USP15 recruitment. We now have added a sentence to include mention of c-Myc as an example of a proteasomal target in the discussion.

Reviewer #3:

This manuscript introduces a well-designed RapTag system that enables inducible proximity between deubiquitylases (DUBs) and their potential substrates using an FKBP-FRB* heterodimerisation strategy. The authors benchmark the platform using USP15 and demonstrate that enforced proximity of USP15 to the immune checkpoint receptor CTLA4 markedly increases CTLA4 stability. Stabilisation is strongly dependent on catalytic activity and on the heterodimeriser, and is accompanied by a substantial elevation of CTLA4 at the plasma membrane. The combination of biochemical, imaging, HiBiT-based quantification, and ubiquitin enrichment assays supports a model in which USP15 recruitment counteracts ubiquitin-dependent lysosomal sorting, thereby promoting CTLA4 recycling.

The study is clearly presented, technically rigorous, and offers a flexible platform for evaluating DUB-substrate interactions with direct implications for the rational design of enhancing chimeras (ENTACs). The work advances an emerging field and has clear conceptual importance for targeted protein rescue strategies.

I find the manuscript strong and suitable for publication after minor revision aimed solely at improving flow and sharpening the central message.

1. Several Results subsections contain rich technical detail, but the overarching conceptual thread, which enforced DUB proximity can redirect protein fate by altering trafficking decisions, occasionally becomes diluted. Can the authors add **brief bridging statements at the end or beginning of relevant subsections** to help readers follow the **logical progression from platform establishment to validation to mechanistic insight to functional outcome**.

Response: We have now added a few additional bridging statements and throughout aimed to improve clarity of transitions. .

2. The observation that inactive USP15 accumulates in intracellular puncta and undergoes reduced steady-state expression is interesting but described briefly. Add **one clarifying sentence in the Results or Discussion explaining the likely mechanism** (e.g., bystander degradation due to complexing with ubiquitylated CTLA4), ensuring the message remains sharp and accessible to non-specialists.

Response: Such a sentence was already included in the result section on page 7:
“We reasoned that this may reflect bystander degradation via rapid shuttling of ubiquitylated CTLA4-USP15 CS heterodimers to the lysosome.”

We now have further edited this statement as suggested by the reviewer:

“We reasoned that this may reflect bystander degradation due to complex formation with CTLA4 and subsequent rapid shuttling of ubiquitylated CTLA4-USP15 CS heterodimers to the lysosome.”

In the discussion, we had also revisited this point on page 10: “...not only is USP15 CS unable to deflect CTLA4 from endolysosomal sorting, it is also itself directly sorted along the lysosomal pathway by virtue of its own association with CTLA4.”

We have changed this to

“...not only is USP15 CS unable to deflect CTLA4 from endolysosomal sorting, it is also itself directly sorted to and degraded in the lysosome by virtue of its own association with CTLA4.”

January 8, 2026

RE: Life Science Alliance Manuscript #LSA-2025-03563R

Prof. Sylvie Urbe
University of Liverpool
Cellular and Molecular Physiology
University of Liverpool
Crown Street
Liverpool, Merseyside L69 3BX
United Kingdom

Dear Dr. Urbe,

Thank you for submitting your Research Article entitled "Targeted recruitment of USP15 enhances CTLA4 surface levels and restricts its degradation." It is a pleasure to let you know that your manuscript is now accepted for publication in Life Science Alliance. Congratulations on this interesting work.

DISTRIBUTION OF MATERIALS:

Again, congratulations on a very nice paper. I hope you found the review process to be constructive and are pleased with how the manuscript was handled editorially. We look forward to future exciting submissions from your lab.

Sincerely,
